# AGNOSTIC LEARNING OF GENERAL ReLU ACTIVATION USING GRADIENT DESCENT

**Pranjal Awasthi**
Google Research
pranjalawasthi@google.com

**Alex Tang**[*]
Northwestern University
alextang@u.northwestern.edu

**Aravindan Vijayaraghavan**[*]
Northwestern University
aravindv@northwestern.edu

## ABSTRACT

We provide a convergence analysis of gradient descent for the problem of agnostically learning a single ReLU function under Gaussian distributions. Unlike prior work that studies the setting of zero bias, we consider the more challenging scenario when the bias of the ReLU function is non-zero. Our main result establishes that starting from random initialization, in a polynomial number of iterations gradient descent outputs, with high probability, a ReLU function that achieves an error that is within a constant factor of the optimal i.e., it is guaranteed to achieve an error of $O(OPT)$, where $OPT$ is the error of the best ReLU function. This is a significant improvement over existing guarantees for gradient descent, which only guarantee error of $O(\sqrt{d \cdot OPT})$ even in the zero-bias case (Frei et al., 2020). We also provide finite sample guarantees, and obtain similar guarantees for a broader class of marginal distributions beyond Gaussians.

## 1 INTRODUCTION

Gradient descent forms the bedrock of modern optimization algorithms for machine learning. Despite a long line of work in understanding and analyzing the gradient descent iterates, there remain several outstanding questions on whether they can provably learn important classes of problems. In this work we study one of the simplest learning problems where the properties of gradient descent are not well understood, namely *agnostic learning of a single ReLU function*.

More formally, let $\tilde{D}$ be a distribution over $\mathbb{R}^d \times \mathbb{R}$. A ReLU function is parameterized by $w = (\tilde{w}, b_w)$ where $\tilde{w} \in \mathbb{R}^d$ and $b_w \in \mathbb{R}$. For notational convenience, we will consider the points to be in $\mathbb{R}^{d+1}$ by appending $\tilde{x}$ with a fixed coordinate 1 as $x = (\tilde{x}, 1)$. Let $D$ be the distribution over $\mathbb{R}^{d+1} \times \mathbb{R}$ induced by $\tilde{D}$. We define the loss incurred at $w = (\tilde{w}, b_w)$ to be

$$L(w) = \frac{1}{2} \mathop{\mathbb{E}}_{(\tilde{x},y) \sim \tilde{D}} \left[ (\sigma(\tilde{w}^\top \tilde{x} + b_w) - y)^2 \right] = \frac{1}{2} \mathop{\mathbb{E}}_{(x,y) \sim D} \left[ (\sigma(w^\top x) - y)^2 \right].$$

Here $\sigma(x) = \max(x, 0)$ is the standard rectified linear unit popularly used in deep learning. The goal in agnostic learning of a ReLU function (or agnostic ReLU regression) is to design a polynomial time learning algorithm that takes as input i.i.d. samples from $D$ and outputs $w = (\tilde{w}, b_w)$ such that $L(w)$ compares favorably with $OPT$ that is given by

$$OPT := \min_{w = (\tilde{w}, b_w) \in H} \frac{1}{2} \mathop{\mathbb{E}}_{(x,y) \sim D} [(\sigma(w^\top x) - y)^2].$$

Here the hypothesis set $H$ that algorithm competes with is the set of ReLU units with parameters $w = (\tilde{w}, b_w)$ with the relative bias $|b_w|/\|\tilde{w}\|_2$ being bounded. This is a non-trivial and interesting regime; when the bias is too large in magnitude the optimal ReLU function fitting the data is either the constant zero function almost everywhere, or a linear function almost everywhere.

This agnostic learning problem has been extensively studied and polynomial time learning algorithms exists for a variety of settings. This includes the noisy teacher setting where $\mathbb{E}[y|x]$ is given by

---
[*]The last two authors are supported by the National Science Foundation (NSF) under Grant No. CCF-1652491 and CCF 1934931. The last author was also funded by a Google Research Scholar award.

a ReLU function Kakade et al. (2011); Mukherjee & Muthukumar (2020) and the fully agnostic setting where no assumption on $y$ is made (Goel & Klivans, 2019; Diakonikolas et al., 2020). In a recent work (Frei et al., 2020) analyzed the properties of gradient descent for the above agnostic learning problem when the bias term is assumed to be zero. The gradient descent based learning algorithm corresponds to the following sequence of updates starting from a suitable initializer $w_0$: $w_{t+1} = w_t - \eta \nabla L(w_t)$. The work of Frei et al. (2020) proved that starting from zero initialization and for distributions where the marginal of $x$ satisfies some mild assumptions , gradient descent iterates produce, in polynomial time, a point $w_T$ such that $L(w_T) = O(\sqrt{OPT})$ when the domain for $x$ is bounded (it is instructive for this bound to think of $OPT < 1$; the general expression is more complicated with some additive terms and dependencies on problem-dependent quantities).

While the above provides the first non-trivial learning guarantees for gradient descent in the case of agnostic ReLU learning, it suffers from a few key limitations. The result of Frei et al. (2020) only applies in the setting when the distribution has a *bounded domain* and when the *bias terms are zero*. When the distribution is not bounded, the error of $O(\sqrt{OPT})$ also includes some dimension-dependent terms; e.g., when the marginal of $\widetilde{x}$ is a standard Gaussian $\mathcal{N}(0, I_{d \times d})$, it gives a $O(\sqrt{d \cdot OPT})$ error. Moreover, there is a natural question of improving the bound of $O(\sqrt{OPT})$ on the error of gradient descent (since the most interesting regime of parameters is when $OPT \ll 1$). This is particularly intriguing given the recent result of Diakonikolas et al. (2020) that shows that, assuming zero bias, gradient descent on a convex surrogate for $L(w)$ achieves $O(OPT)$ error. This raises the question of whether the same holds for gradient descent on $L(w)$ itself. In another recent work, the authors in Vardi et al. (2021) are able to provide convergence guarantees for gradient descent in the presence of bias terms, but under the strong *realizability* assumption, i.e, assuming that $OPT = 0$.

To summarize the existing guarantees, to the best of our knowledge, (i) there are no existing guarantees for any polynomial time algorithm (including gradient descent) for agnostic learning of a ReLU function with bias, and (ii) even in the zero bias case, there is no existing guarantee for gradient descent (on the standard squared loss) that achieves $O(OPT)$ error.

## 1.1 OUR RESULTS

In this work we make progress on both these fronts, by improving the state of the art of guarantees for gradient descent for agnostic ReLU regression. In particular, we show that when the marginal of $x$ is a Gaussian, gradient descent on $L(w)$ achieves an error of $O(OPT)$, even under the presence of bias terms that are bounded. The $O(OPT)$ guarantee that we get even in the zero bias case answers an open question raised in the work of Frei et al. (2020). There are also no additional dependencies on the dimension. Given the recent statistical query lower bound of Goel & Klivans (2019) that rules out an additive guarantee of $OPT + \varepsilon$ for agnostic ReLU regression, our result shows that vanilla gradient descent on the target loss already achieves near optimal error guarantees. Below we state our main theorem. For convenience we assume that $\|\tilde{v}\|_2$ (the optimal weight, i.e. $v = (\widetilde{v}, b_v) \in H$ such that $L(v) = OPT$), is a constant; Appendix C shows why this is without loss of generality.

**Theorem 1.1.** *Let $C_1 \geq 1, C_2 > 0, c_3 > 0$ be absolute constants. Let $D$ be a distribution over $(\widetilde{x}, y) \in \mathbb{R}^d \times \mathbb{R}$ where the marginal over $\widetilde{x}$ is the standard Gaussian $\mathcal{N}(0, I)$. Let $H = \{w = (\widetilde{w}, b_w) : \|\widetilde{w}\| \in [1/C_1, C_1], |b_w| \leq C_2\}$, and consider population gradient descent iterates: $w_{t+1} = w_t - \eta \nabla L(w_t)$. For a suitable constant learning rate $\eta$, when starting from $w_0 = (\widetilde{w}_0, 0)$ where $\widetilde{w}_0$ is randomly initialized from a radially symmetric distribution, with at least constant probability $c_3 > 0$ one of the iterates $w_T$ of gradient descent after $poly(d, \frac{1}{\varepsilon})$ steps satisfies $L(w_T) = O(OPT) + \varepsilon$.*

Please see Section 4 for the more formal statement and proof. Note that the above guarantee applies to one of the intermediate iterates produced by gradient descent within the first $poly(d, 1/\varepsilon)$ iterations. This is consistent with other convergence guarantees for gradient descent in non-realizable settings where last iterate guarantees typically do not exist Frei et al. (2020). One can always pick the iterate among the first $poly(d, 1/\varepsilon)$ steps that has the smallest loss on an independent sample from the distribution $D$.

The above theorem proves that gradient descent obtains a bound of $O(OPT)$ when the relative bias of the optimal ReLU function is bounded (recall that $\|\tilde{v}\|_2 = \Theta(1)$ for the optimal classifier without loss of generality from Proposition C.1). Note that we do not constrain the gradient updates to remain in the set $H$. This result significantly improves upon the existing state-of-the-art guarantees Frei et al.

(2020) of $O(\sqrt{d \cdot OPT})$ for gradient descent even when specialized to the case of ReLU activations with no bias. Further this gives the first provable guarantees in the setting with non-zero bias. Our improved bound of $O(OPT)$ error even with non-zero bias involves several new ideas. At a high level there are two main ingredients that allow us to do beyond the previous work: (1) an improved analysis for gradient descent in the agnostic case that in particular avoids any dimension-dependent factors, and (2) a new "multiscale" random initialization scheme with a stronger guarantee for the initializer. We outline these in more detail in Section 4 and Section 5 respectively.

We remark that some of the assumptions in Theorem 1.1 are made with a view towards a clearer exposition, and similar guarantees hold in more general settings. While the above theorem gives guarantees for gradient descent on the population loss function $L(w)$ (as in Vardi et al. (2021)), we also prove guarantees for the empirical loss function in Section D. Moreover while the above Theorem 1.1 assumes Gaussian marginals (as this already illustrates the improvements guarantees in a basic and well-studied setting), these techniques extend to a broader class of distributions that we describe next.

## 1.2 GUARANTEES BEYOND GAUSSIAN MARGINALS

The above algorithmic result can be generalized to a broader class of marginals than Gaussians, that we call $O(1)$-regular marginals.

$O(1)$-**regular marginals: Assumptions about the marginals over $\widetilde{x}$** We make the following assumptions about the marginal distribution $\widetilde{\mathcal{D}}_x$ over $\widetilde{x} \in \mathbb{R}^d$: there exists absolute constants $\beta_1, \beta_2', \beta_2, \beta_3, \beta_4, \beta_5 > 0$ and $\beta_0 : \mathbb{R}_+ \to \mathbb{R}_+$, such that

(i) Approximate isotropicity and bounded fourth moments: for every unit vector $u \in \mathbb{R}^d$, $\mathbb{E}_{\widetilde{x} \sim \widetilde{\mathcal{D}}_x}[\langle u, \widetilde{x} \rangle^2] \in [1/\beta_2', \beta_2]$, and $\mathbb{E}_{\widetilde{x} \sim \widetilde{\mathcal{D}}_x}[\langle u, \widetilde{x} \rangle^4] \leq \beta_4$.

(ii) Anti-concentration: there exists an absolute constant $\beta_3 > 0$ such that for every unit vector $\tilde{u} \in \mathbb{R}^d$ and $\delta > 0$,

$$\sup_{t \in \mathbb{R}} \mathbb{P}_{\widetilde{x} \sim \widetilde{\mathcal{D}}_x} \left[ \langle \tilde{u}, \widetilde{x} \rangle \in (t - \delta, t + \delta) \right] \leq \min\{\beta_3 \delta, 1\}.$$

(iii) Spread out: there exists $\beta_0 : \mathbb{R}_+ \to \mathbb{R}_+$ such that $\beta_0(|b_v|) > 0$ is a constant when $|b_v|$ is a constant, and

$$\forall \tilde{v} \in \mathbf{S}^{d-1}, \quad \mathbb{E}_{\widetilde{x} \sim \widetilde{\mathcal{D}}_x} \left[ \sigma(\tilde{v}^\top \widetilde{x} + b_v) \right] \geq \beta_0(|b_v|).$$

(iv) 2-D projections: In every 2-dimensional subspace of $\mathbb{R}^d$ spanned by orthonormal unit vectors $\tilde{u}_1, \tilde{u}_2 \in \mathbb{R}^d$, we have a set $G_{\tilde{u}_1, \tilde{u}_2} \subset \mathbb{R}$ such that ,

$$\mathbb{P}_{\widetilde{x} \sim \widetilde{\mathcal{D}}_x} \left[ \tilde{u}_2^\top \widetilde{x} \in G_{\tilde{u}_1, \tilde{u}_2} \right] = 1 - o(1), \text{ and} \tag{1}$$

$$\forall t \in G_{\tilde{u}_1, \tilde{u}_2}, \quad \mathbb{E}_{\widetilde{x} \sim \widetilde{\mathcal{D}}_x} \left[ \sigma(\tilde{u}_1^\top \widetilde{x}) \mid \tilde{u}_2^\top \widetilde{x} = t \right] \geq \beta_5 \cdot \mathbb{E}_{\widetilde{x} \sim \widetilde{\mathcal{D}}_x} \left[ \sigma(\tilde{u}_1^\top \widetilde{x}) \right]. \tag{2}$$

In other words, the conditional expectation of $\sigma(\tilde{u}_1^\top \widetilde{x})$ is not much smaller after conditioning on the projection in an orthogonal direction $\tilde{u}_2$, for most values of $\tilde{u}_2^\top \widetilde{x}$. Note that for a Gaussian $N(0, I)$, the r.v.s $\tilde{u}_1^\top \widetilde{x}, \tilde{u}_2^\top \widetilde{x}$ are independent, so this condition holds with $\beta_5 = 1$ and $G_{\tilde{u}_1, \tilde{u}_2} = \mathbb{R}$.

We remark that Gaussian distribution $\mathcal{N}(0, I)$ is $O(1)$-regular i.e., all the constants $\beta_1, \beta_2, \beta_2', \beta_5 = 1, \beta_3 \leq 2$, and $\beta_0(b_v) = \mathbb{E}_{g \sim N(0,1)}[\sigma(g + b_v)] > 0$ for all $b_v \in (-\infty, \infty)$; in fact $\beta_0$ is an increasing function that is 0 only at $-\infty$. We also note that assumptions of this flavor have also been used in prior works including Vardi et al. (2021), which inspired parts of our analysis. In particular, Vardi et al. (2021) assume a lower-bound on the density for any 2-dimensional marginal; our assumption (4) on the 2-dimensional marginals is qualitatively weaker (it is potentially satisfied by even discrete distributions), and moreover we only need the condition to be satisfied for a large fraction of values of $\tilde{u}_2^\top \widetilde{x}$ (and not all). See Section B for the generalized version of our main theorem.

## 2 RELATED WORK

The agnostic ReLU regression problem that we consider has been studied in a variety of settings. In the *realizable* setting or when the noise is stochastic with zero mean, i.e., $\mathbb{E}[y|x]$ is a ReLU function, the learning problem is known as isotonic regression and can be solved efficiently via the GLM-tron algorithm (Kakade et al., 2011; Kalai & Sastry, 2009). Distributions generated by a 1-layer ReLU neural network under the realizable setting can also be learned efficiently (Wu et al., 2019). In the absence of any assumptions on the distribution of $y|x$, the work of Goel & Klivans (2019) provided an efficient algorithm that achieves $O(OPT^{2/3}) + \varepsilon$ error under Gaussian and log-concave marginals in the zero-bias setting. The authors also show that it is hard to achieve an additive bound of $OPT + \varepsilon$ via statistical query (SQ) algorithms Kearns & Valiant (1994). For the case of zero bias and any marginal over the unit sphere, the work of Goel et al. (2017) provides agnostic learning algorithms for the ReLU regression problem that run in time exponential in $1/\varepsilon$ and achieve an error bound of $OPT + \varepsilon$. The recent work of Diakonikolas et al. (2020) improved the upper bound of Goel & Klivans (2019) to $O(OPT) + \varepsilon$ via designing an efficient algorithm that performs gradient descent on a convex surrogate for the loss $L(w)$; very recently they also obtained near optimal sample complexity with a regularized loss (Diakonikolas et al., 2022). Note that all of the above works that study the fully agnostic setting consider the setting where the bias terms are not present.

Recent works of Frei et al. (2020); Vardi et al. (2021) consider analyzing gradient descent for the ReLU regression problem. Frei et al. (2020) provides an $O(\sqrt{OPT})$ guarantee (along with some additional problem-dependent terms) for the case of zero bias and bounded distributions. When considering distributions such as the standard Gaussian $\mathcal{N}(0, I)$ the bound of Frei et al. (2020) incurs a dimension dependent term of the form $O(\sqrt{d} \cdot \sqrt{OPT})$ in the error bound. Vardi et al. (2021) provides a tighter analysis that also extends to the case of non-zero bias. However the analysis only applies in the realizable setting, i.e., when $OPT$ is zero. Our main result provides improved bounds over these works by providing a dimension independent error bound that applies to the case of non-zero bias as well.

There is also a long line of work analyzing gradient descent for broader settings. The works of Ge et al. (2015; 2018); Jin et al. (2017); Anandkumar & Ge (2016); Soltanolkotabi (2017) show convergence of gradient descent updates to approximate stationary points in non-convex settings under suitable assumptions on the function being optimized. Another line of work considers the global convergence properties of gradient descent. These works establish that gradient descent on highly overparameterized neural networks converges to the global optimum of the empirical loss over a finite set of data points (Allen-Zhu et al., 2019; Du et al., 2019; Jacot et al., 2021; Zhong et al., 2017; Chizat & Bach, 2018; Lee et al., 2019; Arora et al., 2019). Yet another line of work considers the realizable setting where data is generated from an unknown small depth and width neural network. These works analyze the local convergence properties of gradient descent when starting from a suitably close initial point (Bartlett et al., 2018; Zou et al., 2020).

## 3 PRELIMINARIES

We consider agnostically learning a single ReLU neuron with bias through gradient descent under the supervised learning setting. We assume we are given data $(x, y)$, where $x \in \mathbb{R}^{d+1}$ follows the standard Gaussian distribution $\mathcal{N}(0, I)$ in the first $d$ dimensions and the $d + 1$'th dimension being a constant 1. We also assume the labels $y \in \mathbb{R}$ are arbitrarily correlated with $x$ and $\sigma(w^\top x)$.

Note that throughout the paper, we will use $\widetilde{w}, \widetilde{v}, \widetilde{x}$ to denote the first $d$ dimensions of $w, v, x$ respectively, with the last dimension of $w$ being $b_w \in \mathbb{R}$ (similarly for $b_v \in \mathbb{R}$). Therefore, $w^\top x$ is in fact $\widetilde{w}^\top \widetilde{x} + b_w$.

In the analysis, we will compare the current iterate $w$ to any optimizer of the loss $L(w)$.

$$v := \arg\min_{w \in H} L(w), \text{ where } L(w) = \frac{1}{2} \mathbb{E}_{(x,y) \sim D} \left[ (\sigma(w^\top x) - y)^2 \right], \tag{3}$$

and the hypothesis set $H = \{w = (\widetilde{w}, b_w) : \|\widetilde{w}\| \in [\frac{1}{C_1}, C_1], |b_w| \leq C_2)\}$, where $C_1$ and $C_2$ are absolute constants. This is to ensure that the relative bias $|b_w|/\|\widetilde{w}\|_2$ is bounded; as described earlier Appendix C allows us to assume $\|\widetilde{w}\| \in [\frac{1}{C_1}, C_1]$ without loss of generality.

As we are in the agnostic setting, there may be no $w$ that achieves zero loss. We can split the loss function $L(w)$ into two components, one of which is $F(w)$ defined by

$$F(w) \coloneqq \frac{1}{2} \mathbb{E}\left[(\sigma(w^\top x) - \sigma(v^\top x))^2\right], \quad \nabla F(w) \coloneqq \mathbb{E}\left[(\sigma(w^\top x) - \sigma(v^\top x))\sigma'(w^\top x)x\right]. \quad (4)$$

We will often refer to $F(w)$ as the *realizable loss*, since it captures the difference between $w$ and $v$; in the realizable setting $L(w) = F(w)$. Note that $F(v) = 0$.

**Gradient of the Loss.** The gradient of $L(w)$ with respect to $w$ is

$$\nabla L(w) = \mathbb{E}\left[(\sigma(w^\top x) - y)\sigma'(w^\top x)x\right] \quad (5)$$

where $\sigma'(\cdot)$ is the derivative of $\sigma(\cdot)$, defined as $\sigma'(z) = \mathbb{1}\{z \geq 0\}$. Note that the ReLU function $\sigma(z)$ is differentiable everywhere except at $z = 0$. Following standard convention in this literature, we define $\sigma'(0) = 1$. Note that the exact value of $\sigma'(0)$ will have no effect on our results.

We can also decompose $\nabla L(w)$ as

$$\nabla L(w) = \mathbb{E}\left[(\sigma(w^\top x) - \sigma(v^\top x))\sigma'(w^\top x)x\right] + \mathbb{E}\left[(\sigma(v^\top x) - y)\sigma'(w^\top x)x\right] \quad (6)$$

Therefore, $\nabla L(w) = \nabla F(w) + \mathbb{E}\left[(\sigma(v^\top x) - y)\sigma'(w^\top x)x\right] \quad (7)$

**Gradient Descent.** Finally, our paper focuses on the standard gradient descent algorithm with a fixed learning rate $\eta > 0$. We initialize at some point $w_0 \in \mathbb{R}^{d+1}$, and at each iteration $t \in \mathbb{N}$ we have $w_{t+1} = w_t - \eta \nabla F(w_t)$. We do not optimize the iteration count in this paper; hence it will be instructive to think of $\eta$ as a non-negligible parameter that can be set to be sufficiently small (e.g., an inverse polynomial for polynomial time guarantees).

**Simplification.** For sake of exposition we will assume that $\|\tilde{v}\|_2 = 1$; the same analysis goes through when $\|\tilde{v}\|_2 \in [1/C_1, C_1]$ as well. Moreover Proposition C.1 shows that assuming that $\|\tilde{v}\|_2$ is normalized is without loss of generality. Note that we cannot make such a simplifying assumption about the vectors $w_t = (\tilde{w}_t, b_w)$ in the intermediate iterations.

Finally, please see Section B for the weaker distributional guarantees and guarantees.

## 4 OVERVIEW OF THE ANALYSIS (PROOF OF THEOREM 1.1)

We now provide an overview of our analysis. For complete proofs of the lemmas and propositions, please refer to the supplementary material (Appendix A). Recall that our goal throughout the learning process is to find a $w \in \mathbb{R}^{d+1}$ such that $L(w)$ achieves a comparable performance to $OPT = L(v)$. In order to accomplish this, we aim to find $w$ such that it is close to $v$, i.e. $\|w - v\|$ is small. Note that approximating $v$ suffices to achieve an error close to $OPT$, since we can upper-bound $L(w)$ as

$$L(w) = \frac{1}{2}\mathbb{E}\left[(\sigma(w^\top x) - y)^2\right] = \frac{1}{2}\mathbb{E}\left[(\sigma(w^\top x) - \sigma(v^\top x) + \sigma(v^\top x) - y)^2\right]$$

$$\leq 2 \cdot \frac{1}{2}\mathbb{E}\left[(\sigma(w^\top x) - \sigma(v^\top x))^2\right] + 2 \cdot \frac{1}{2}\mathbb{E}\left[(\sigma(v^\top x) - y)^2\right] = 2F(w) + 2OPT$$

through Young's inequality. The realizable portion of the loss $F(w)$ becomes $O(OPT)$ when $\|w - v\| \leq O(\sqrt{OPT})$ (see Lemma 4.4 for a proof), and as a consequence we will get $O(OPT)$ error in total.

To formalize our intuition above, we adopt a similar proof strategy used in Frei et al. (2020). Namely, we argue that when optimizing with respect to the agnostic loss $L(w_t)$, we are always making some non-trivial progress due to a decrease in $\|w_t - v\|$ and due to a decrease in $F(w_t)$ (which is just the realizable portion of the loss). Moreover, whenever we stop making progress, we will argue that at this point either $\|w_t - v\| \leq O(\sqrt{OPT})$ or $\|\nabla F(w_t)\| \leq O(\sqrt{OPT})$; in both cases, this iterate already achieves an error of $O(OPT)$ due to Lemma 4.4 and Lemma 4.3.

**Challenges in arguing progress.** At a high-level the analysis of gradient descent follows a similar approach to Frei et al. (2020) which only handles zero bias. Yet there are several new ideas needed to obtain the stronger $O(OPT)$ guarantee even for the zero-bias case. Moreover, allowing non-zero bias terms imposes extra technical challenges. For example, the probability measure of $\{w^\top x \geq 0, v^\top x \geq 0\}$ under Gaussian distributions, which is vital to deriving the gain in each gradient descent step, does not have a closed-form expression when bias is present. Furthermore we cannot afford to lose any dimension dependent factors or assume boundedness. Thus, to address these difficulties, more detailed analyses (e.g. Lemma 4.1, 4.2) are needed to facilitate our argument.

Moreover tackling non-zero bias terms requires additional assumptions when initializing $w_0$ as well. The initializer finds a $w_0$ such that $F(w_0)$ is strictly less than $F(0)$ by a constant amount $\delta > 0$ (this is inspired by Vardi et al. (2021), however $\delta$ in their case can have an inverse-polynomial dependence on the dimension ). In fact our multiscale random initialization and the improved analysis is crucial to obtaining a dimension-independent bound on the error. The high-level intuition behind why this property is useful is that it ensures that gradient descent does not get trapped around a highly non-smooth region (e.g. when $w = 0$) by making it start at somewhere better than it, so that $w$ keeps moving closer to $v$. Moreover, in our case the analysis is more challenging to implement compared to Vardi et al. (2021) because of the agnostic setting. This is because Vardi et al. (2021) heavily relies on the realizability assumption to simplify its analysis.

We also highlight our improvements on the dependency of the dimension $d$. In previous works, the guarantees of the algorithm has a dependence on $d$ either explicitly or implicitly. For instance, in Frei et al. (2020) the $O(\sqrt{OPT})$ guarantee for ReLU neurons includes a coefficient in terms of $B_X$ (the upper-bound for $\|x\|$), which for Gaussian inputs is in fact $\sqrt{d}$; or for example in Vardi et al. (2021), the gain for each gradient descent iteration $\gamma$ comes with a dependency on $c$ (the upper-bound for $\|x\|$) of $c^{-8}$, which for Gaussian is $d^{-4}$. In contrast, we avoid such dependencies on the dimension $d$ in order to obtain our guarantees.

We first establish two important lemmas we will later utilize in proving progress in each iteration. As stated in the preliminaries, we assume in the rest of the section that $\|\tilde{v}\|_2 = 1$. The first lemma gives a lower bound on the measure of the region where both $\sigma(v^\top x)$ and $\sigma(w_t^\top x)$ are non-zero. Our inductive hypotheses will ensure that this lower bound is a constant (if $|b_v|$ is a constant).

**Lemma 4.1** (Lower bound on the measure of the intersection). *Suppose the marginal distribution $\widetilde{\mathcal{D}}_x$ over $\widetilde{x}$ is $O(1)$-regular. There exists an absolute constant $c > 0$ such that for all $\delta > 0$, if $F(w) \leq F(0) - \delta$ then*

$$\mathbb{P}[w^\top x \geq 0, v^\top x \geq 0] \geq \frac{\delta^2}{c\|w\|_2^4\|v\|_2^4} = \frac{\delta^2}{c\|w\|_2^4(1 + |b_v|^2)^2}. \tag{8}$$

With Lemma 4.1, the following lemma allows us to get an improvement on the realizable portion of the loss function as long as the gradient is non-negligible. We state and prove this lemma for the general case of $O(1)$-regular marginal distributions.

**Lemma 4.2** (Improvement from the first order term). *Suppose the marginal over $\widetilde{x}$ is $O(1)$-regular. There exists absolute constants $c_1, c_2 > 0$ such that for any $\delta > 0$, if $\|v\|_2, \|w\|_2 \leq B$ and $F(w) \leq F(0) - \delta$, then $\langle \nabla F(w), w - v \rangle \geq \gamma\|w - v\|^2$, where $\gamma = \frac{c_1\delta^9}{B^{28}}$.*

The constants $c_1, c_2$ depend on the constants $\beta_1, \beta_2', \beta_2, \beta_4$ etc. in the regularity assumption of $\widetilde{\mathcal{D}}_x$. We remark that for our setting of parameters $\delta = \Omega(1)$ and $B = O(1)$, and hence we will conclude that $\langle \nabla F, w - v \rangle \geq \Omega(\|w - v\|_2^2)$. Please refer to Appendix A for all the complete proofs.

## 4.1 MAIN PROOF STRATEGY

With these two key lemmas, we are now ready to discuss the proof overview of the main theorem (Theorem 1.1). We inductively maintain two invariants in every iteration of the algorithm:

$$\text{(A)} \quad \|w_t - v\|_2 \leq O(1), \quad \text{and} \quad \text{(B)} \quad F(0) - F(w_t) = \Omega(1).$$

These two invariants are true at $t = 0$ due to our initialization $w_0$. Lemma B.3 guarantees with at least constant probability $\Omega(1)$, both the invariants hold for $w_0$. The proof that both the invariants

continue to hold follows from the progress made by the algorithm due to a decrease in both $\|w_t - v\|_2$ and $F(w_t)$ (note that we only need to show they do not increase to maintain the invariant).

The argument consists of two parts. First, assuming $F(w_t) \leq F(0) - \delta$ holds (for some constant $\delta > 0$), we establish that whenever $\|w_t - v\|^2 > \gamma OPT$ for some constant $\gamma > 0$, gradient descent always makes progress i.e. $\|w_t - v\|^2 - \|w_{t+1} - v\|^2$ is lower bounded. Next, we argue that if $w_0$ is initialized such that $F(w_0) \leq F(0) - \delta$ for some constant $\delta > 0$, then throughout gradient descent $F(w_t)$ always decreases, i.e. the inequality $F(w_t) \leq F(w_0) \leq F(0) - \delta$ always holds.

However, unlike Vardi et al. (2021) where they focus on the realizable setting, analyzing gradient descent on the agnostic loss $L(w)$ is more challenging, since the update depends on $\nabla L(w)$ and not $\nabla F(w)$. In fact, the additional term from the "non-realizable" portion of the loss $L(w)$ can overwhelm the contribution from the realizable loss when either $\|\nabla F\|_2 \leq O(\sqrt{OPT})$ or $\|w_t - v\|_2 \leq O(\sqrt{OPT})$. The following two lemmas argue that in both of these cases, the current iterate already achieves $O(OPT)$ error (and this iterate will be the $T$ that satisfies the guarantee of Theorem 1.1).

**Lemma 4.3** (Success if $\|\nabla F\| \leq O(\sqrt{OPT})$). *Suppose $B, \delta > 0$ are constants such that $\|v\|_2, \|w\|_2 \leq B$ and $F(w) \leq F(0) - \delta$. Then there exists a constant $C_G > 0$, such that if $\|\nabla F(w)\| \leq C_G \sqrt{OPT}$ then $\|w - v\|_2 \leq O(\sqrt{OPT})$.*

*Proof.* We can first apply Lemma 4.2 to conclude that $\langle \nabla F(w), w - v \rangle \geq \gamma \|w - v\|^2$ for some constant $\gamma > 0$ (since $B, \delta > 0$ are constants), hence we have $\|\nabla F(w)\| \|w - v\| \geq \langle \nabla F(w), w - v \rangle \geq \gamma \|w - v\|^2$. Thus $\|w - v\|_2 = O(\sqrt{OPT})$ which implies the lemma. $\square$

We now argue that if $\|w_t - v\| \leq O(\sqrt{OPT})$, then $F(w_t) \leq O(OPT)$ through the following lemma; this is stated and proven for $O(1)$-regular distributions.

**Lemma 4.4** (Small $\|w_t - v\|$ implies small $F(w_t)$). *Assume $\widetilde{\mathcal{D}}_x$ is $O(1)$-regular with parameters defined above. If $\|w_t - v\|_2 \leq O(\sqrt{OPT + \varepsilon})$ for some $\varepsilon > 0$, then $F(w_t) \leq O(OPT + \varepsilon)$.*

*Proof.* Since ReLU function is 1-Lipschitz (i.e. $|\sigma(z) - \sigma(z')| \leq |z - z'|$),

$$F(w_t) = \frac{1}{2} \mathbb{E}\left[(\sigma(w_t^\top x) - \sigma(v^\top x))^2\right] \leq \frac{1}{2} \mathbb{E}\left[(w_t^\top x - v^\top x)^2\right] = \frac{\|w_t - v\|^2}{2} \mathbb{E}\left[(u^\top x)^2\right]$$

where we defined $u = \frac{w_t - v}{\|w_t - v\|}$, hence the last equation. Now, notice by using Young's inequality, we get

$$\mathbb{E}\left[(u^\top x)^2\right] = \mathbb{E}\left[(\widetilde{u}^\top \widetilde{x} + b_u)^2\right] \leq 2\mathbb{E}\left[(\widetilde{u}^\top \widetilde{x})^2\right] + 2b_u^2 \leq 2\beta_2 + 2b_u^2 \leq O(1)$$

due to the regularity assumption on $\widetilde{\mathcal{D}}_x$. Hence

$$F(w_t) \leq \frac{\|w_t - v\|^2}{2} \cdot O(1) \leq O(\|w_t - v\|_2^2) \leq O(OPT + \varepsilon)$$

which concludes the proof. $\square$

**Proving progress in $\|w_t - v\|$ and $F(w_t)$.** To show $\|w_t - v\|$ decreases, we establish the following lemma.

**Lemma 4.5** (Decrease in $\|w_t - v\|$). *Assume at time $t$, $F(w_t) \leq F(0) - \delta$ where $\delta > 0$ is a constant and $\widetilde{\mathcal{D}}_x$ is $O(1)$-regular. For constants $\eta = \frac{0.05 \cdot \gamma}{d\beta_2}, C_p = \frac{1}{9}\left(\sqrt{\frac{100\beta_2^2/\gamma^2 + 90}{\beta_2/\gamma}} + 10\sqrt{\frac{\beta_2}{\gamma}}\right), C' = 19.8\gamma/\beta_2$ where $\gamma$ is defined as in Lemma 4.2, if for some $\varepsilon > 0$ $\|w_t - v\|^2 > \gamma^{-1}C_p^2(OPT + \varepsilon)$, then $\|w_{t+1} - v\|^2 \leq \|w_t - v\|^2 - \eta C'(OPT + \varepsilon)$.*

As a direct consequence of Lemma 4.5, we obtain the following inductive statement: for every $t$, either (a) $\|w_t - v\|^2 - \|w_{t+1} - v\|^2 \geq \eta C(OPT + \varepsilon)$ is true for some constant $C > 0$ or (b) $\|w_t - v\|^2 \leq O(\gamma^{-1}(OPT + \varepsilon))$ holds. Observe that when (b) holds Lemma 4.4 implies the loss is $O(OPT)$; hence we need only assume at time $t$ (b) does not hold yet, thus it suffices focusing on showing (a) is true. Additionally, note at each timestep $t$,

$$\|w_t - v\|^2 - \|w_{t+1} - v\|^2 = 2\eta \langle \nabla L(w_t), w_t - v \rangle - \eta^2 \|\nabla L(w_t)\|^2$$

Therefore, to lower-bound $\|w_t - v\|^2 - \|w_{t+1} - v\|^2$, we will give a lower bound for $\langle \nabla L(w_t), w_t - v \rangle$ and an upper bound for $\|\nabla L(w_t)\|^2$. To show that $F(w_t)$ decreases we show that at time $t$, if gradient descent continues to make progress towards $v$, then $F(w_{t+1}) \leq F(w_t) \leq F(0) - \delta$. The progress in $F(w)$ follows crucially relies on Lemma 4.2. Please see Appendix A in the supplementary material for the detailed proofs.

## 5 RANDOM INITIALIZATION

We now prove the initialization lemma assuming weak conditions on the marginal distribution over $\widetilde{x} \in \mathbb{R}^d$ which is $\widetilde{\mathcal{D}}_x$ (recall that the standard Gaussian $N(0, I)$ also satisfies all of the properties). We will initialize $w = (\tilde{w}, b_w)$ with $b_w = 0$ and $\tilde{w}$ drawn from a spherical symmetric distribution $\mathcal{D}_w$. The length is chosen from the distribution $\mathcal{D}_\rho$ so that it has a non-negligible probability in any constant length interval $(a_1\|v\|_2, a_2\|v\|_2)$ where $a_2 > a_1 > 0$ are constants: our specific choice picks the correct length scale with non-negligible probability, and is reasonably spread out.

Our new random initialization and the improved analysis are crucial in obtaining the $O(OPT)$ guarantee even with non-zero bias. Our *multiscale random initialization* scheme tries out different length scales and ensures that with non-negligible probability we get an initializer that satisfies the required property. For the correct guess of length scale of $\|\tilde{v}\|_2$ (up to a factor of 2), our improved analysis (see (10)) shows that the random spherically symmetric initialization with constant probability produces an initializer $w$ with $F(w) - F(0) = -\Omega(\|\tilde{v}\|_2^2)$. When we have unknown length scale $\|\tilde{v}\|_2 \in [1/M, M]$, the random initialization can try out the different length scales in geometric progression i.e., the length scale $\tau$ is chosen uniformly at random from $\{2^{-j} : j \in \mathbb{Z}, -\log M \leq j \leq \log M\}$.

**Multiscale random initialization** We are given a parameter $M$ such that $\|v\|_2 \in [2^{-\log M}, 2^{\log M}]$ (note that $M$ can have large dependencies on $d$ and other parameters; our guarantees will be polynomial in $\log M$). A random initializer $w = (\tilde{w}, 0)$ is drawn from $\mathcal{D}_{\text{unknown}}(M)$ as follows:

1. Pick $j$ uniformly at random from $\{-\lceil\log M\rceil, -\lceil\log M\rceil + 1, \ldots, -1, 0, 1, \ldots, \lceil\log M\rceil\}$.

2. $\rho \in \mathbb{R}_+$ is drawn according to $\mathcal{D}_\rho$ as follows: we first pick[1] $g \sim N(0, 1)$ and set $\rho = 2^j|g|$.

3. A uniformly random *unit* vector $\hat{w} \in \mathbb{R}^d$ is drawn and we output $\tilde{w} = \rho\hat{w}$. The initializer is $(\tilde{w}, 0)$.

We prove the following claim about the multiscale random initializer.

**Lemma 5.1.** *There exists $c_1(v), c_2(v), c_3(v) > 0$ which only depend on $b_v/\|\tilde{v}\|_2$ (and not on the dimension), and are both absolute constants when $|b_v|/\|\tilde{v}\|_2 = O(1)$, such that the following holds. When $w = (\tilde{w}, b_w = 0)$ is drawn according to the distribution $\mathcal{D}_{unknown}(M)$ described above for some given $M \geq 1$ satisfying $\|v\|_2 \in [1/M, M]$. Then with probability at least $c_2(v)/\log M$,*

$$F(w) \leq F(0) - c_1(v)^2\|\tilde{v}\|_2^2, \text{ and } \|w - v\| \leq c_3(v)\|\tilde{v}\|_2 \tag{9}$$

In the above lemma, if $\widetilde{\mathcal{D}}_x$ is a standard Gaussian $N(0, I)$, the descriptions of these above constants become much simpler, as described in Section B.3. The guarantees for the multiscale random initialization scheme follows from the analysis of random initialization when the length scale of $\|\tilde{v}\|_2 = 1$ is known. Without loss of generality (see Section C, we can assume that $\|\tilde{v}\|_2 = 1$ (or $\Theta(1)$). For convenience, we will set $\mathcal{D}_\rho$ to be the absolute value of a standard Gaussian $N(0, 1)$ (or $N(0, \beta^2)$ with $\beta \in [1, 2]$. In this setting, we can show for constants $c_1(v), c_2(v), c_3(v) > 0$ (these are constants when $|b_v|/\|\tilde{v}\|_2$ is bounded), we have with probability at least $c_2(v) > 0$

$$F(w) \leq F(0) - c_1(v)^2\|\tilde{v}\|_2^2, \text{ and } \|w - v\| \leq c_3(v)\|\tilde{v}\|_2. \tag{10}$$

We remark that for random initialization to work, we only need the probability of success $\eta \geq c_2(v) > 0$ to be non-negligible (e.g., at least an inverse polynomial). We can try $O(1/\eta)$ many random initializers, and amplify the success probability to be very close to 1.

---

[1]One can pick many other spread out distributions in place of the absolute value of a Gaussian.

**Overview of the proof of Lemma 5.1** We now outline the argument of Lemma 5.1. Please refer to Section B.3 and Section B.4 for the full proofs. For convenience we define $\widehat{b}_v := b_v/\|\tilde{v}\|_2, \widehat{v} := v/\|\tilde{v}\|_2$, so they are normalized w.r.t. the length of $\tilde{v}$. The conditions of the lemma assume that $|\widehat{b}_v| = O(1)$. The multiscale random initialization finds the correct length scale with probability at least $1/(\log M)$. For the rest of the overview we assume that the length $\|\tilde{v}\|_2 = 1$ is known; without loss of generality (see Section C), we can assume that $\|\tilde{v}\|_2 = 1$. By definition, the distribution of $\tilde{w} \in \mathbb{R}^d$ is spherically symmetric.

$$F(w) - F(0) = \frac{1}{2} \mathbb{E}_x \left[ (\sigma(\tilde{w}^\top x) - \sigma(\tilde{v}^\top x + b_v))^2 \right] - \frac{1}{2} \mathbb{E}_x \left[ \sigma(\tilde{v}^\top x + b_v))^2 \right]$$

$$= \frac{\rho^2 \|\tilde{v}\|_2^2}{2} \mathbb{E}_x \left[ (\sigma(\widehat{w}^\top x)^2 \right] - \rho \|\tilde{v}\|_2^2 \mathbb{E}_x \left[ \sigma(\widehat{w}^\top x)\sigma(\widehat{v}^\top x + \widehat{b}_v)) \right],$$

where $\tilde{w} = \rho \|\tilde{v}\|_2 \widehat{w}$ with $\widehat{w}$ being the unit vector along $\tilde{w}$. For a fixed $\rho \in \mathbb{R}_+$, $\widehat{w}$ (and hence $\tilde{w}$) is picked along a uniformly random direction i.e., $\widehat{w} \sim_U \mathbb{S}^{d-1}$. Hence for $x \sim \widetilde{\mathcal{D}}_x$,

$$\mathbb{E}_{\widehat{w} \sim \mathbb{S}^{d-1}}[F((\rho\widehat{w}, 0)) - F(0)] = \frac{\rho^2 \|\tilde{v}\|_2^2}{2} \mathbb{E}_{\widehat{w} \sim_U \mathbb{S}^{d-1}} \mathbb{E}_{x \sim \widetilde{\mathcal{D}}_x} \left[ (\sigma(\widehat{w}^\top x)^2 \right] \quad (11)$$

$$- \rho \|\tilde{v}\|_2^2 \mathbb{E}_{\widehat{w} \sim_U \mathbb{S}^{d-1}} \mathbb{E}_{x \sim \widetilde{\mathcal{D}}_x} \left[ \sigma(\widehat{w}^\top x)\sigma(\widehat{v}^\top x + \widehat{b}_v)) \right] = \|\tilde{v}\|_2^2 (c'\rho^2 - 2c_3(v)\rho)$$

where $c' > 0$ is a universal constant based on our assumptions about $\widetilde{\mathcal{D}}_x$ ($c' = 0.5$ for $x \sim N(0, I)$). One technical portion of the argument is to derive an expression for $c_3(v)$, and prove that it is a constant independent of the dimension. This forms the bulk of the argument and requires symmetrization and careful use of anti-concentration bounds. Once we establish this, we need to prove that the first part (10) holds with non-negligible probability. From (11), we note that for any $\rho \in \left[\frac{c_3(v)}{2c'}, \frac{c_3(v)}{c'}\right]$, we have that

$$\mathbb{E}_{\widehat{w} \sim_U \mathbb{S}^{d-1}}[F((\rho\widehat{w}, 0))] \leq F(0) - \|\tilde{v}\|_2^2 \frac{c_3(v)^2}{2c'}.$$

Moreover $\rho$ is distributed as the absolute value of a standard normal with variance in $[1, 4]$; so we get from anti-concentration bounds that $\rho$ is in the right interval with probability at least $c_5(v) > 0$, which is constant when $|\widehat{b}_v|$ is a constant. Now we condition on this event that $\rho \in \left[\frac{c_3(v)}{2c'}, \frac{c_3(v)}{c'}\right]$. For a fixed $\rho$ in this interval, let $Z$ be a r.v. that captures the distribution of $F((\rho\|\tilde{v}\|\widehat{w}, 0)) - F(0)$ as $\widehat{w}$ is drawn uniformly from the unit sphere $\mathbb{S}^{d-1}$. Note that $\mathbb{E}[Z] \leq -\|\tilde{v}\|_2^2 c_3(v)^2/2c'$.

$$\text{Var}[Z] \leq \mathbb{E}[F((\rho\|\tilde{v}\|_2\widehat{w}, 0))^2] \leq O(1) \cdot \|\tilde{v}\|^4 \left(2\beta_4 + \widehat{b}_v^4\right).$$

Further for $\lambda = -\mathbb{E}[Z]/2$, we have from the Cantelli-Chebychev one-sided tail inequality we have for some absolute constant $c_6 > 0$

$$\mathbb{P}\left[Z \leq \mathbb{E}[Z]/2\right] \geq \frac{\mathbb{E}[Z]^2}{4\text{Var}[Z] + \mathbb{E}[Z]^2} \geq \min\left\{c_6 c_3(v)^2/(\beta_4 + \widehat{b}_v^4), \frac{1}{2}\right\} =: c_6(v),$$

where $c_6(v)$ is a constant when $\widehat{b}_v$ is a constant. This allows us to conclude that $F(w) < F(0) - \Omega(\|\tilde{v}\|^2)$ with probability at least $c_5(v) \cdot c_6(v)$ which is a constant when $\widehat{b}_v$ is a constant. Finally $\|w - v\|_2 \leq \|w\|_2 + \|\tilde{v}\|_2$ is upper bounded just because of our choice of $\rho$ and $\|\tilde{v}\|_2$ being upper bounded by assumption. See Sections B.3 and B.4 for the full proofs.

## 6 CONCLUSION

In this paper, we provided a convergence analysis of gradient descent for learning a single neuron with general ReLU activations (with non-zero bias terms) and gave improved guarantees under comparable assumptions also made in previous works. We addressed multiple challenges for analyzing general ReLU activations with non-zero bias terms throughout our analyses that may lead to better understanding of the dynamics of gradient descent when learning ReLU neurons. However, our analysis does not apply to modern neural networks that have multiple nodes and layers. The major open direction is to generalize current performance guarantees for networks of multiple neurons and higher depth.

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
