# OpenReview forum: "Agnostic Learning of General ReLU Activation Using Gradient Descent"
_ICLR.cc/2023/Conference — ICLR 2023 poster_

### Official Review · Reviewer_DGFB · 2022-10-18

**Confidence:** 4
**Correctness:** 3
**Technical Novelty And Significance:** 4
**Empirical Novelty And Significance:** Not applicable
**Recommendation:** 8

**Clarity, Quality, Novelty And Reproducibility:**

The paper is written in a clear style that is not difficult to follow. I doubt the quality of the paper because of the problems found above. There is a novelty since this paper claims better convergence analysis. Since this paper studies the theoretical behavior of algorithms, reproducibility is not applied.

**Details Of Ethics Concerns:**

The paper studies machine learning theory, and there is no ethics concern.

**Strength And Weaknesses:**

Strength:
1. the introduction of O(1) regularity
2. the better convergence results

Weakness:
1. For Lemma 4.3, please specify the limit process for the big-O notation.
2. When we read Section 1.2, a natural question is whether the O(1) regular marginal conditions can be induced from a sufficient condition: the distribution Dx has a continuous and compactly supported density function, which takes a positive value at the origin? Whether, subject to scaling x|-> ax for a specific a>0 because of (iii), this density condition is also necessary? If not, would the O(1) regularity contain any meaningful application scenario that is excluded from the C0 density assumption?
3. For the simplification right before Section 4, would the analysis be different if the optimizer \tilde{v} appears at the boundary of H?
4. Please provide the Appendix as cited in the paper. For the time being, we are using the arXiv version of this paper to review the Appendix, and the comments are made accordingly.
5. Please specify the limit process for the notations \Omega(1) and O(1), below Equation (9). The same problem is in (1), and at the beginning of Section 4.1.
6. Since in the proof Lemma 4.2 is cited, please add the O(1) regularity assumption to Lemma 4.3
7. The constant c2 plays no role in Lemma 4.2. I guess this constant can be removed.
8. For the arXiv version (dated August 04, 2022, same as below). The last inequality of (11) of the arXiv version: the O(1) distribution is replaced by the Normal distribution. The same mistake is observed in Equation (20) of the arXiv version.
10. The O(1) regularity assumption just assumes beta_4 is positive, which could be very small. So, in the last two paragraphs, from beta<q^(1/4)/beta_4 one can not derive |b_u|>1/2.
11. The claim (page 10 of the arXiv version) "Observe that when (b) holds Lemma 4.4 implies the loss is O(OPT)" requires the specification of the definition of "O(OPT)", to be further checked. Also, why would this lead to Lemma 4.5?
12. Lemma 4.6 (the arXiv version) is different from the cited source. In Lemma d.4 in [VYS21], the underlying distribution is assumed compactly supported, and the bound c of the distribution support is changed to the square root of d (space dimension) in Lemma 4.6. In this paper, the O(1) regularity does not guarantee compact support. I guess that such a difference may not affect the conclusion, but then the new lemma should be proved separately.
13. The O(1) regularity should be assumed for Lemma 4.5, since in the proof Lemma 4.2 is employed. Also, the boundedness of w_t and v should either be proved or assumed.
14. Line -3 on page 11 of the arXiv version: according to Assumption (i) in O(1) regularity, the bound should be beta_2, not 2.
15. Line 9 on page 12 of the arXiv version: there should be a factor $\sqrt{beta_2}$ on the right-hand side.
16. Line -4 of page 12 of the arXiv version: if C_2 is larger than C_1 C_p^2, then C' is negative.
17. Line 3 of page 15 of the arXiv version: C_G is not elsewhere claimed to be greater than sqrt(2).
18. Line -5 of page 6 of the ICLR version: the existence of such w_t (in particular, w_0) should be assumed. In particular, if v=0, such existence is violated.

**Summary Of The Paper:**

This paper studies the behavior of a single ReLU neuron in the gradient descent training process. The main results include that first, a new class of distributions (called O(1)-regular marginals) is proposed to characterize the regularity of the input space distributions. This class includes multivariate Gaussian distributions, and does not require compact support. Second, under the proposed O(1)-regularity, and with constant step-size, Theorem 1.1 claims that in polynomial (of the input space dimension, and the error threshold) number of iterations, the expected loss converges to the optimal value subject to a small error threshold.

======================

update after the discussion session: we find that most of our concerns have been resolved. We are therefore happy to recommend acceptance of this paper.

**Summary Of The Review:**

There are some mathematical mistakes. Appendix is claimed but missing.

---

> ### Author Response · Authors · 2022-11-18
> **Thanking Reviewer DGFB for Their Feedback**
>
> We thank the reviewer for their detailed feedback. We appreciate all the suggestions and minor corrections that were brought up – this will help polish our writeup more. We would like to respectfully counter a couple of points:
> (i) We did attach an appendix even in the original submission – this was in the .zip file as part of the Supplementary material – this is how we were instructed to submit the Appendix/ Supplementary material. (This was also present before the rebuttal phase – see e.g., the review by Reviewer na9i which mentions the Appendix D.) We feel it’s unfair to penalize us for following the conference policy.
> (ii) Some of the concerns raised are about the use of the Big-Oh notation, which is standard in CS. In many of the cases, we use this notation to avoid carrying around constants that are cumbersome and may make the statements harder to read. The order of quantifiers is clear from the context. However, we will clarify this explicitly where they may be doubt (especially in places where the reviewer has raised this point).
> (iii) Finally,  we emphasize that none of these points affect the correctness of our result or its argument.
>
> We address some of the reviewer’s specific concerns (using the same numbering) in the next comment box due to limitation on characters.

---

> > ### Author Response · Authors · 2022-11-18
> > **Thanking Reviewer DGFB for Their Feedback - Cont'd**
> >
> > We address some of the reviewer’s specific concerns (using the same numbering) below:
> >
> > 1. See about Big-Oh notation above.
> > 2. It is possible that there is a simpler to state (but not as general) sufficient condition. However, a condition on just the density at 0 (+ compactness) does not suffice. Firstly, note that since we are dealing with non-zero bias, we need anti-concentration and spread around points/offsets that are non-zero. This is not captured by the density condition at 0. On the other hand, a lower bound on the density is only meaningful for constant dimensional marginals. Also, note that even for a Gaussian a density lower bound at 0 alone does not suffice e.g., when $\sigma^2$ is very small (tending to 0), the density at 0 becomes very large, but it does not have any spread or anti-concentration far from 0; in fact for non-zero bias, it may not give much signal about $\tilde{v}$.  Furthermore, our conditions can potentially capture even discrete distributions e.g., a discrete Gaussian (which is used in many settings, including differential privacy, and even in hardness of learning problems) satisfies all of these conditions.
> > 3. It does not matter even if $\tilde{v}$ appears in the boundary of $H$. One way to see why it doesn’t matter is that we can also get the same bounds by changing $H$ so that $||v||$ is in a slightly expanded interval $[1/(2C_1), 2 C_1)]$ – the analysis produces the same guarantee with slightly worse constants.  Also, note that our algorithm does not use knowledge of the choice of $H$ (it only needs an upper bound on the length $||\tilde{v}||$). Finally, as Proposition C.1 shows, it is essentially  scale invariant w.r.t. $||\tilde{v}||$.
> > 4. As stated earlier, we did include the appendix in the supplementary materials section in the OpenReview panel.
> > 5. See about Big-Oh notation above.
> > 6. We will add the pre-condition into the lemma statement.
> > 7. Thank you for catching the typo: we will remove $c_2$ as suggested.
> > 8. We thank the reviewer for pointing out this typo. We will update it accordingly
> > 9. We remark that one can always assume without loss of generality that $\beta_4$ is a sufficiently large constant. This is because any distribution that has a fourth moment bounded by $\beta_4$, also has its fourth moment bounded by $\beta’_4$ for any $\beta’_4 > \beta_4$.  Hence we can just apply the argument with a $\beta_4$ that is certainly larger than $(q/2)^4$ (note that this is upper bounded by a constant; in fact, a reasonably small one). But we thank you for bringing this up. We do not mention an explicit condition on $\beta_4$ since it makes the theorem statement less understandable, and as we discussed we can assume this without loss of generality. We will clarify this in the paper, and mention quite early on that we can assume without loss of generality that $\beta_4$ can be thought of as being sufficiently large constant.
> > 10. Thank you for pointing out this grammatical error: It was supposed to say “As a direct consequence of Lemma 4.5, we get the following inductive statement…”. We fixed it now. In the final version, in the informal description at the bottom of page 7, we will specify that $\gamma$ is a constant for our purposes (or just remove $\gamma$ altogether in the informal description).
> > 11. Thank you for bringing this to our attention. We acknowledge that this lemma from [DVY22] assumes compactness (though it uses it in a very weak sense) –  but the same argument essentially goes through even without this condition, even in our setting (see Lemma A.1 in Appendix A). Moreover note the bounds from this lemma (the second order term) do not affect the error guarantee of $O(OPT)$ – they only affect the iteration count by placing a bound on the step size $\eta$.
> > 12. We will add the assumption on the distribution accordingly. While $v$ is bounded by assumption, we will also state explicitly that $w_t$ is also bounded due to our assumption that $F(w_t)$ is bounded.
> > 13. Thanks for catching the typo.
> > 14. Thanks for catching the typo.
> > 15. We will explicitly clarify the order of choosing the constants in the final version (see the updated Lem. 4.5 in the ICLR version). In particular our constant $C_2$ is chosen so that it is much smaller than $C_1 C_p^2/2$
> > 16. Again, we will explicitly specify how the constants are chosen at the beginning of the proof of Theorem 1.1 (including the constant $C_G$); note that it is chosen only at this point. Please see the updated Appendix A (Proof of Theorem 1.1) in the ICLR version for how the constants are specified.
> > 17. We guarantee that the condition holds for $w_0$ because of the initialization step. For the informal description/ proof outline, we are focussing on a successful outcome of the random initialization step (we prove this holds w.h.p. for at least one of the random initializers in Lemma 5.1). Also note that $v=0$ does not satisfy the conditions of Theorem 1.1.

---

> > > ### Comment · Reviewer_DGFB · 2022-11-29
> > > **most concerns well resolved**
> > >
> > > I apologize for not checking carefully the review system and missing the supplementary materials.
> > >
> > > Q6. The O(1) regularity assumption is not added to Lemma 4.3 in the updated version.
> > >
> > > Q7. The constant c2 is not removed from Lemma 4.2.
> > >
> > > Q9. The discussion happens at line 3 of page 14 in the appendix (the page and line numbers are of the updated "Supplementary Material"). First, the authors agree with me that the condition $(q/2)^{1/4}/(4\beta_4)\leq 1/2$ is introduced on the fly, for proving that $|b_u|>1/2$. Second, I do not agree with the proposed solution that $\beta_4$ be claimed to be sufficiently large. First, such claim, if made, should be declared clearly as needed, as that $\beta_4 > (q/2)^{1/4} / 8$. Otherwise, this is a mistake. Second, the idea that $\beta_4$ can be taken arbitrarily large free of price, is wrong. This is because at the end of the proof of Lemma 4.2, $1/\beta_4$ (which is therefore a very small number) is used to bound a power of $\delta$. So, the authors need to claim in Lemma 4.2, that instead of "any" such delta, one needs to make sure that the delta does not exceed a power of $1/\beta_4$.
> > >
> > > Q11. We see that the authors moved Lemma 4.6 into Lemma A.1 and provide proof. Due to time constraints, and also that some arguments in "Lemma D.4 of Vardi et al. (2021)" are cited, we do not double-check the proof.
> > >
> > > Among these points, Q6 and Q9 concern mistakes, While other points are minor and I am confident that readers can identify them easily. We believe that the authors can easily fix these problems. We note that this paper represents solid research results. We are therefore happy to recommend acceptance of this paper. The grades have been updated.

---

### Official Review · Reviewer_na9i · 2022-10-24

**Confidence:** 3
**Correctness:** 3
**Technical Novelty And Significance:** 3
**Empirical Novelty And Significance:** Not applicable
**Recommendation:** 6

**Clarity, Quality, Novelty And Reproducibility:**

The paper is written quite clearly in general and is easy to follow. The quality of the result is good as it improves upon the existing work of (Frei et al. 2020). There seem to be new ideas used in the analysis for gradient descent along with a novel initialization scheme which are to my knowledge novel.

**Strength And Weaknesses:**

Pros: The paper has the following strengths.

1. The exposition is in general clear and easy to follow. The problem setup and related work are detailed nicely that makes it convenient for the reader.

2. The analysis is an improvement over that of (Frei et al. 2020) since this latter work assumes zero-bias ReLU's and achieves an error bound of $O(\sqrt{d \ OPT})$ when the samples are drawn from a Gaussian distribution.

Cons:

1. I believe that the related work is incomplete for gradient descent learning of ReLU functions. The work of Soltanolkotabi [1] considers this problem in the realizable setting (with zero-bias) with the inputs drawn from a Gaussian and derives error bounds for recovering the unknown parameter. It is not clear to me how easy it is to extend this analysis to the noisy teacher setting. Since [1] is well cited, it would be good to check other related works that cite [1] in order to be more complete in the literature review.

2. I find it is a bit restrictive to analyze gradient descent on the population risk. While this is natural to do as a first step in the analysis and certainly non-trivial, I believe the main theoretical result should ultimately be the one for the finite sample setting and should ideally be presented in the main text. This analysis is currently relegated to Appendix D as Theorem D.1. I also have a question regarding Theorem D.1 later below.

[1] M. Soltanolkotabi, Learning relus via gradient descent, NeurIPS 2017.

Further comments:

1. In the statement of Theorem D.1, it is not clear to me how $|y| \leq B_Y$ can be assumed since the input $\tilde{x}$ is generated from a Gaussian. For instance, isn't this violated in the realizable setting where y is defined as a ReLU?

2. I was trying to understand whether in the special case of the "noisy teacher" setting (or "realizable" setting) the analysis says something about the convergence of the gradient descent iterates to the ground truth parameter. However from Section 4.1, it seems to me that this is not the case since it is only mentioned that the iterates lie within a ball of radius $O(\sqrt{OPT})$ around the ground truth parameter. But we know from [1] that iterates of projected gradient descent (on the empirical loss) converge linearly to the ground truth in the realizable setting.  Some more explanation in this regard would be helpful.

3. In Section 1.1 just above Theorem 1.1, $\tilde{v}$ is introduced but has not been defined till then. There is a small typo towards the bottom of page 3 in "Vardi et al."


**Summary Of The Paper:**

The paper studies the problem of agnostically learning a ReLU function via gradient descent using samples drawn from Gaussian (or even a more general class of) distributions.  The main result states that under a suitable random initialization, if gradient descent is run for sufficiently many iterations on the population risk loss function, then it outputs a ReLU function the error for which is within a constant factor of the optimal error, i.e., the minimum value of the population risk (referred to as: OPT). This result does not necessarily need the bias of the ReLU to be zero and improves existing guarantees for gradient descent (Frei et al. 2020).

**Summary Of The Review:**

I think the contributions of the paper are decent and improve upon the analysis of (Frei et al. 2020) for learning ReLUs. The paper is also written cleanly and is easy to follow. As mentioned under "weaknesses" and in "further comments", I do have some comments regarding some closely related work and other technical aspects which need clarification.

---

> ### Author Response · Authors · 2022-11-18
> **Thanking Reviewer na9i for Their Feedback**
>
> We thank the reviewer for their feedback. We will include the suggested paper into the discussion of related works.
>
> Regarding presenting Theorem D.1 as the main theorem: We chose to present the population result (Theorem 1.1) as the main result since this result captures the main challenges and difficulty in the agnostic setting. By focussing on the population setting in the main body, we can convey the main new ideas that lead to the result. The analysis extends to the empirical version using fairly straightforward techniques – so we deferred this to the Appendix. For the final version, we can move the statement of Theorem D.1 to the main body. We remark that the population setting is already interesting — in fact, Vardi et al. (NeurIPS’21) on learning a realizable ReLU activation focused exclusively on the population setting.
>
> To address the comments:
>
> Note that some condition on $y$ is necessary for the sample complexity since there is no distributional assumption on $y$ (and can be unbounded otherwise).  However, we just need the boundedness to hold almost surely, and this bound only features as a weak dependence in the sample complexity (for example, an assumption of sub-Gaussian would also suffice).
>
> In our work we consider gradient descent without projection, whereas in [1] the proof of convergence heavily relies on projecting the iterates onto the subspace such that certain regularization constraints are met, therefore we believe our results are incomparable. However we will include relevant discussions in the related work section and we thank the reviewer for pointing this out.
>
> We thank the reviewer for pointing out the typo. We will fix it in the updated version.

---

### Official Review · Reviewer_4WPP · 2022-10-30

**Confidence:** 3
**Correctness:** 4
**Technical Novelty And Significance:** 3
**Empirical Novelty And Significance:** Not applicable
**Recommendation:** 6

**Clarity, Quality, Novelty And Reproducibility:**

The paper is well-written and easy to follow for the reviewer. Though there’re minor aspects that could be fixed, e.g., equation (14) and (16) are both over length.

The technical novelty could be better highlighted in the analysis. For example, how the analysis incorporate the bias and improve the approximation factor of the optimum value could be better emphasized during the proofs.

**Strength And Weaknesses:**

Strength:
- The result of this paper improves upon a previous result for learning a single ReLU function by additionally incorporating the bias in the analysis. Further, the analysis shows that the convergent iterate is within a constant factor of the optimum, whereas prior result requires an additional factor of d (i.e., input dimension).

Weakness:
- The feature distribution is assumed to be drawn from a Gaussian distribution. While this can be relaxed to the case of (1)-regular marginals, generalizing further seems challenging (e.g., does the condition hold for mixture of Gaussians or anisotropic Gaussians?)
- There are related works that consider the realizable setting of learning ReLU networks, which should perhaps also be discussed in the related work, e.g., here’s one:

“Learning Distributions Generated by One-Layer ReLU Networks.” Shanshan Wu, Alexandros G. Dimakis, Sujay Sanghavi. 2019.

**Summary Of The Paper:**

This paper provides a convergence analysis of gradient descent for learning a single ReLU function under Gaussian feature distributions. The paper focuses on the mean squared loss between the single ReLU function the label, but allows updating both the weight vector and the bias parameter of the ReLU function.

The main contribution is proving that the gradient descent algorithm starting from a random initialization converges to within a constant factor of the global minimum of the loss within polynomial (in input dimension) number of iterations.

Besides the case of Gaussian feature distributions, the analysis can also be generalized to the O(1)-regular marginals.

**Summary Of The Review:**

Overall I think this is a solid paper that contributes to the learning of nonlinear ReLU functions. It expands prior result and clearly states the limitations too (i.e., limited to one node) for future work. As noted above there is some concerns related to missing several related works in the literature.

---

> ### Author Response · Authors · 2022-11-18
> **Thanking Reviewer 4WPP for Their Feedback**
>
> We thank the reviewer for their feedback. We will fix the overflown equations and emphasize the technical novelty in the analyses.  We will also add the relevant paper into related works as suggested.
>
> While generalizing to more distributions is an interesting question, we would like to remark that the Gaussian marginals case is already of great theoretical interest. In fact most theoretical algorithmic guarantees for ReLU activation (including the realizable single-layer ReLU network setting) make this assumption. Moreover we generalize to a broader class of distributions. Regarding the question of anisotropic distributions, our conditions do capture anisotropic distributions with well-conditioned covariance matrices i.e., the ratio of largest to the smallest eigenvalue is bounded (otherwise $\tilde{v}$ could be along the badly conditioned direction). It can also capture mixtures of Gaussians, whose means are not too far from the origin (otherwise it would be an issue for existing techniques even in the case of a ReLU with zero-bias).

---

### Official Review · Reviewer_iUj1 · 2022-11-28

**Confidence:** 5
**Correctness:** 4
**Technical Novelty And Significance:** 2
**Empirical Novelty And Significance:** 2
**Recommendation:** 5

**Clarity, Quality, Novelty And Reproducibility:**

The work is sound and well-written.

p.9 has a broken reference.

**Strength And Weaknesses:**

Pros:
1. This is an important problem. Previous results have focused on the unbiased case, so this is somehow the first result for biased ReLU activations.
2. The gradient descent happens in bias and the weight vector, which means that the naive idea of extending the dimension works

Cons:
1.  The authors state in the abstract that this is the first algorithm based on gradient descent that achieves these guarantees when the bias is zero. But [3,4] apply gradient descent and achieve $O(OPT)$. The first work does gradient descent on a surrogate loss which differs from this work, where the gradient descent is applied to the standard objective. Meanwhile, the second work uses gradient descent on the same objective and achieves $O(OPT)$. So, I believe this sentence should be removed from the abstract.
2.  This work considers the case where the ratio between $||w||_2$ and $b_w$ is bounded by a constant. The title and the abstract leave the impression that this work provides a theorem for any value of bias which is not the case. One of the main difficulties of learning ReLU activations is to sample enough points so that $\sigma'(w^Tx+b)>0$  and $\sigma(u^Tx+b')>>0$ ($u,b'$ are the best parameters for this instance) for a large portion of the points (see [5] theorem 4.2 for the intuition); with these assumptions, this is somehow easily satisfied (for standard Gaussian) with a good initialization (Section 5) because the mean and variance of the RV $u^Tx+b'$ is bounded by constants.




[3]: Ilias Diakonikolas, Surbhi Goel, Sushrut Karmalkar, Adam R. Klivans, Mahdi Soltanolkotabi. Approximation Schemes for ReLU Regression
[4]: Ilias Diakonikolas, Vasilis Kontonis, Christos Tzamos, and Nikos Zarifis. Learning a single neuron with adversarial label noise via gradient descent
[5]: Gilad Yehudai, and Ohad Shamir. Learning a Single Neuron with Gradient Methods


**Summary Of The Paper:**

This work studies the convergence of gradient descent in the regime of learning ReLU activations agnostically for a class of distributions that contains the standard high dimensional normal distribution. In contrast with previous results, they consider the case where bias exists and apply gradient descent on the weight vector+bias. Their approach essentially connects the analysis of [1] and [2].

[1]: Spencer Frei, Yuan Cao, and Quanquan Gu. Agnostic learning of a single neuron with gradient descent.
[2]: Gal Vardi, Gilad Yehudai, and Ohad Shamir. Learning a single neuron with bias using gradient descent

**Summary Of The Review:**

I think the contributions of this work are not sufficient for acceptance in ICRL.

---

> ### Author Response · Authors · 2022-11-30
> **Response to Reviewer iUj1**
>
> We thank the reviewer for their feedback. We were unaware of this very recent work [4] which also gives an O(OPT) guarantee for zero bias (on the standard L_2^2 objective). We remark that our work was obtained independently of [4] (which was put up on the ArXiv on June 17, 2022). However, we will remove the statement in the abstract about improving upon existing work even in the zero-bias case, since this is not the main result of our paper. We will also  update the discussion regarding the related work [4] in our paper.
>
> Regarding the second question about the constant bias regime, we will clarify in the abstract that our guarantees are not for general biases. However, we respectfully push back about how the ideas in [5, Theorem 4.2] may imply our result. The ideas in [5] are for the realizable case, and further incur several dimension dependent terms even for Gaussian marginals (which are too costly for O(OPT) guarantees). Please see  Section 4 (Challenges in arguing progress), page 6 for a more detailed description.

---

### Decision · Program_Chairs · 2023-01-20

**Decision:**

Accept: poster

**Justification For Why Not Higher Score:**

The main disadvantage of the paper is that the main result relies on a strong assumption. In particular, the assumption that the ratio between the 2-norm of the weight vector and the threshold is bounded by a constant is critical to the analysis. It is not clear that the ideas here will contribute to extending the result beyond this assumption.


**Justification For Why Not Lower Score:**

The contribution is reasonable and extends an important line of work.


**Metareview: Summary, Strengths And Weaknesses:**

This paper studies the problem of learning generalized linear models with a ReLU activation in the presence of adversarial label noise.
Their main result is that gradient descent on the natural $L_2^2$ objective efficiently converges to a solution with error within a constant factor from optimal, under Gaussian marginals (or slightly weaker distributional assumptions). What distinguishes this paper from prior work is that they can handle some non-centered ReLU activations, namely functions of the form $ReLU(w \cdot x+t)$, for some non-zero values of the threshold $t$. Most prior work had studied the case $t=0$.

A disadvantage of this work is that the algorithm/analysis does not cover "general ReLU activations", as stated in the abstract. As pointed out by the reviewers, the paper considers the case where the ratio between the 2-norm of the weight vector and the threshold is bounded by a constant. This is a non-trivial restriction but the work still makes a contribution to this line of research.



**Note From Pc:**

if the above contains the word "oral" or "spotlight" please see: "oral" presentation means -> notable-top-5% and "spotlight" means -> notable-top-25%. As stated in our emails, we are disassociating presentation type from AC recommendations